# Enhancing the Impact of Genomics Research in Autism through Integration of Research Results into Routine Care Pathways—A Case Series

**DOI:** 10.3390/jpm11080755

**Published:** 2021-07-30

**Authors:** Iskra Peltekova, Daniela Buhas, Lara Stern, Emily Kirby, Afiqah Yusuf, Mayada Elsabbagh

**Affiliations:** 1McGill University Health Centre, McGill University, Montreal, QC H3A 0G4, Canada; daniela.buhas@mcgill.ca (D.B.); lara.stern@mcgill.ca (L.S.); mayada.elsabbagh@mcgill.ca (M.E.); 2Montreal Children’s Hospital, McGill University Health Centre, Montreal, QC M4A 3J1, Canada; 3Centre of Genomics and Policy, McGill University, Montreal, QC H3A 0G1, Canada; emily.kirby@mcgill.ca; 4Azrieli Centre for Autism Research, Montreal Neurological Institute, Montreal, QC H3A 2B4, Canada; afiqah.yusuf@mail.mcgill.ca

**Keywords:** return of genetic research results, autism spectrum disorder, neurodevelopmental conditions, clinical care pathways

## Abstract

The return of genetic results (RoR) to participants, enrolled as children, in autism research remains a complex process. Existing recommendations offer limited guidance on the use of genetic research results for clinical care. We highlight current challenges with RoR and illustrate how the use of a guiding framework drawn from existing literature facilitates RoR and the clinical integration of genetic research results. We report a case series (*n* = 16) involving the return of genetic results to participants in large genomics studies in Autism Spectrum Disorders (ASD). We outline the framework that guided RoR and facilitated integration into clinical care pathways. We highlight specific cases to illustrate challenges that were, or could have been, resolved through this framework. The case series demonstrates the ethical, clinical and practical difficulties of RoR in ASD genomic studies for participants enrolled as children. Challenges were resolved using pre-established framework to guide RoR and incorporate research genetic results into clinical care. We suggest that optimal use of genetic research results relies on their integration into individualized care pathways for participants. We offer a framework that attempts to bridge the gap between research and healthcare in ASD.

## 1. Introduction

The rapid advancement of genomic research in Autism Spectrum Disorders (ASD) has been supported by the growing participation of affected children and families, and by the use of increasingly powerful genomic tests, such as microarrays, exome and genome sequencing (ES, GS). In the clinic, genetic tests are used to find an etiology for the behaviorally defined diagnosis of ASD, which may guide healthcare for the affected individual or family [1,2]. However, existing clinical genetic testing recommendations are outdated and do not take into account the large increase in genetic information about ASD [3]. They also do not consider the variability in care pathways within different healthcare systems [4]. The genetic and clinical heterogeneity of ASD makes the return of genetic results (RoR) complex, requiring individualized genetic counseling and health management [5]. Recommendations by the American College of Medical Genetics and Genomics only offer direction on RoR related to secondary findings from clinical genetic testing [6]. Existing clinical RoR recommendations lag behind the fast pace of genomic discovery from ASD research and there is limited guidance in the context of neurodevelopmental conditions. This may limit access of affected individuals and families to novel genetic information that has the potential to alter their healthcare [7].

When used in research, results from genetic tests like ES and GS accelerate the discovery of new genetic variants associated with ASD but are subject to more reporting challenges than clinical tests [8]. The preferences of research participants with regards to receiving their personal or their child’s genetic results vary, where many are interested while some are not [9]. Researchers may have the responsibility to return genetic results to participants, when these lead to changes in the participants’ healthcare [10]. There are several recommendations on the ethical obligations of researchers for RoR in the research context [11,12]. There are also numerous discussions regarding the broader process of RoR from research, including ethical considerations, scientific validity and clinical applicability of research results, and criteria for RoR to individuals. Table 1 provides a summary of the literature on these topics. However, there are no specific recommendations on which research genetic results should be communicated and how to integrate research genetic results into participants’ clinical care. Furthermore, approaches may differ across different health systems and institutions, which precludes a one-size-fits-all approach [13]. Research teams may rely on clinical recommendations, which are limited in scope [14]. This lack of practical guidance for RoR from genetic research may result in variable practices by research teams [15,16].

Conventional ethics frameworks for RoR are faced with new challenges stemming from the increasing frequency and complexity of genetic findings from genomic technologies. RoR from genetic studies typically aims to return findings that are actionable for participants (i.e., the finding can guide clinical decision-making) [17]. Although this goal is underscored in various recommendations (Table 1), none have offered a comprehensive roadmap for RoR. In this report, we present 16 cases involving the return of actionable genetic research results to participants who enrolled as children, and their families, in large-scale genomic studies in ASD, across two different healthcare jurisdictions in Canada. We highlight specific challenges with the return of research genetic results to participants. We present the framework that guided the individualized RoR process (Figure 1), with the aim to integrate actionable research genetic results into clinical care pathways for participants. We describe five cases in greater detail to illustrate special challenges with the RoR process that were, or could have been, resolved with the implementation of our framework. Our goal is not to offer guidance to research teams on determining actionability or making a decision to proceed with RoR. Instead, our aim is to outline the specific considerations that must be addressed during the RoR process in order to ensure that actionable genetic findings contribute to care in a meaningful way for participants for whom an actionable result is identified. We suggest that achieving optimal utility of genetic research results for participants relies on their integration in routine clinical care pathways. This approach would ensure that research genetic results are maximally utilized, while minimizing potential harm to participants. It may also foster collaboration between research and clinical settings that may contribute to improved interpretation of complex genetic results, up-to-date information on actionable findings, improved care pathways for affected individuals, and accelerated translation of research into clinical care.

## 2. Methods

The cases involved children and their families, who participated in large-scale multi-site genomic studies in ASD between 2007 and 2017, namely, the Simons Simplex Collection (SSC) (www.sfari.org/resource/simons-simplex-collection/; accessed on 20 May 2020) and MSSNG (www.mss.ng; accessed on 20 May 2020), by enrolling through our research site. The SSC recruited children with a clinically determined diagnosis of ASD, their siblings, and biological parents (16). MSSNG continues to recruit children with a clinically determined diagnosis of ASD, as well as siblings and biological parents (17). For both studies, children with ASD and co-morbid conditions (such as intellectual disability) were accepted for enrolment. Families were recruited from hospital or community clinics where children underwent diagnostic assessments. At enrollment, consent was provided by a parent/legal guardian on behalf of the child, and assent was sought from any minor able to provide it. Detailed phenotypic information about the proband and family was collected at the time of enrollment. Genetic analyses were carried out in designated genetic research laboratories at the primary site as part of the respective research study, with the main research technology at the time being microarrays, followed by some exome sequencing towards the end of the research period reported by our study (16, 17, 18). Genetic specialists and experts working with the primary site determined which research genetic results would be considered actionable and important to communicate to the secondary site (us), based on available scientific and phenotypic data in the literature regarding the genetic finding, and its relevance to ASD and related neurodevelopmental conditions. Once an actionable research genetic result was available for a participant, the primary study site initiated RoR by contacting us (secondary site) to complete the process. The secondary research site (us) then combined the participant’s clinical and phenotypic information to finalize the determination of “actionability”. Research team members at our site who assisted with this process included a developmental pediatrician, a clinical geneticist, the principal investigator of the secondary site, and a research assistant, with consultation from an ethicist. There was a time lag between recruitment and RoR, due to lengthy research analyses. Clinical and phenotypic information was obtained by clinical and research charts review. The studies involving human participants were reviewed and approved by the Research Ethics Board of the Research-Institute of McGill University Health Centre. The participants or their legal guardians, if enrolled as children, had provided their written informed consent to participate in the study.

Prior to commencing RoR, we convened a workgroup of local experts from Montreal Children’s Hospital and McGill University to develop a site-specific protocol for RoR to research participants. The workgroup consisted of a geneticist, a child and adolescent psychiatrist, a developmental pediatrician, and a researcher (site investigator in the multi-site study). Ongoing consultation was sought from an ethicist. By reviewing cases as they arose and the existing literature (Table 1), the workgroup iteratively developed the proposed framework presented in Figure 1. The goal was to develop an RoR framework that provided guidance on key aspects when an actionable research genetic result was available, with the intent of result integration into the participant’s clinical care pathway, irrespective of healthcare jurisdiction. The framework involves five principles to guide RoR (Figure 1) on a case-by-case basis:*Relevance of genetic result to current, and future care:* genetic and personal health information should be synthesized to determine or confirm if the research result is actionable.*Participant/family expectations, preferences and decisions*: preferences for receipt of research results should be elicited from the individual/family, at the time of consent and when an actionable result is available.*Person/family-centeredness:* the research team should collaborate with the Most Responsible Clinician (MRC) (primary care or specialist) for the individual/family receiving the genetic result, to foster personalized healthcare pathways.*Care coordination*: routine health services (e.g., access to a genetic specialist) should be actively engaged to ensure that resulting care pathways are clear and accessible.*Benefits and risks*: potential positive and negative impacts of the genetic result on the participant/family and on clinical care pathways should be considered and managed.

## 3. Results

Our research site received genetic research results for 16 participants in total, who enrolled as children, out of 400 enrolled participants. The average time between enrollment and genetic result availability was 5.5 years. Efforts were made to contact families by all means available. Three participants (cases 7, 11, and 16) lost to follow-up and could not be re-contacted. In two cases, genetic results were returned to the family prior to RoR by our site: for case 3, the result was returned by another research study that the family participated in; for case 10, the result was identified on routine clinical genetic testing. Characteristics of the participants and relevant genetic information are summarized in Table 2. All were probands except for one, who was an unaffected sibling. Five of the participants had reached adulthood by the time of RoR. The caregivers who provided consent at enrollment were contacted to facilitate contact with the now-adult participants. Of the genetic results, 12 were CNVs, 4 were SNVs and 1 was aneuploidy. Most genetic changes were on chromosomes 1, 15, and 16. Of the CNV results, 6 were deletions (at 9q21.13, 15q11.2, 15q13.1, Xp22.31, and two at 16p13.11) and 6 were duplications (two at 1q21.1, 15q13.1, and 16p11.2; one at 1q43). The majority of genetic changes (*n* = 10) occurred *de novo*. In the following section, we outline five cases in greater detail in order to highlight specific challenges in the RoR process and illustrate the application of our guiding framework to resolve or circumvent these.

### 3.1. Case 3

JD is a child with ASD and intellectual impairment, who was enrolled in the genetic study at age four. JD was subsequently enrolled in a second genetic study based in a different country. The second research study identified an ES result showing a *de novo* missense SNV in the *Phosphatase and Tensin Homolog* (*PTEN*) gene, mutations in which are associated with hamartoma tumor syndromes (MIM 607028) [18]. The *PTEN* hamartoma tumor syndrome is associated with macrocephaly, developmental delays, and autism [18]. The variant was classified as pathogenic. Its health implications made it clinically actionable and necessitated disclosure.

The second research site, which was in a different healthcare jurisdiction than the family, communicated the result in a letter to the participant’s mother, before the RoR process from the first research site where the family enrolled (our site). The mother was encouraged to seek help from local health services but access to those was not facilitated. The family’s preferences about the management of the research genetic result were not elicited. The family had no guidance on accessing and navigating clinical services within their jurisdiction. The family contacted several researchers and clinicians outside their circle of care to seek guidance. After significant delays, the family obtained access to clinical genetic counseling in their region. 

The case illustrates the negative impacts of RoR arising from the absence of a clear pathway for the integration of actionable research genetic findings into clinical care. This contributed to family distress, delays in service provision, and inefficient use of healthcare resources. The application of our guiding framework (specifically, Principles 2 through 5) could have potentially circumvented these issues, by clarifying family preferences and expectations with regards to the return of the genetic research result, ensuring the RoR approach was family-centered, facilitating care coordination, and monitoring the risks from RoR.

### 3.2. Case 4

JM is a healthy adolescent whose adult sibling has ASD. As children, the siblings enrolled in the genetic study, along with their parents. Research microarray and ES analyses were performed for all. The ES for JM revealed a *de novo* missense SNV in the *Tuberous Sclerosis 2* (*TSC2*) gene. Mutations in *TSC2* may cause Tuberous Sclerosis Complex (TSC) (MIM 191092), an autosomal dominant disorder characterized by hamartomas in several organ systems [19]. TSC is associated with developmental and learning difficulties, central nervous system tumors, and renal problems [19]. At the time of enrollment, JM was a healthy child, with no neurologic or developmental difficulties. The variant in JM was not previously reported in the literature. The central study site reported the finding as a variant of uncertain clinical significance (VUS), possibly pathogenic. The predicted clinical impact was deemed actionable, warranting disclosure to the participant. Clear health surveillance guidelines for TSC exist [20], along with specialized clinics in JM’s community. 

JM’s mother was contacted, as she consented to JM’s research participation at enrollment, to inform her that a genetic result was available. JM’s preferences regarding the receipt of the result were also obtained. JM expressed a desire to learn about the genetic finding. With her permission, the research team collaborated with her MRC (family doctor) on the integration of the research result into JM’s healthcare. The MRC referred JM to her local genetics clinic for confirmatory clinical genetic testing and counseling. The case demonstrates the integration of novel genetic information from research into the clinical care of a research participant, by considering the actionability of findings (Principle 1) and preferences of the participant (Principle 2). Collaboration between research and clinical services resulted in clear and person-centered healthcare pathways (Principle 3 and 4). 

### 3.3. Cases 8 and 9

NM and DM are siblings who have ASD. They enrolled with their parents in the genetic research study. A few years later, during a clinical work-up of NM, a maternally inherited deletion at 15q11.2 was found. The parents were informed of the result by the ordering clinician but were unable to obtain genetic counseling. Soon after, research microarray results became available for both siblings (while still in childhood) from the genetic study: NM had the previously identified maternally inherited deletion; DM had a maternally inherited duplication at 15q13.1 and *de novo* duplication at 1q43. The mother did not have any neurodevelopmental conditions.

Deletions in the 15q11 region have been associated with developmental and neurologic issues, with variable penetrance and expressivity [21]. The deletion at 15q11.2 was deemed a VUS by the research laboratory. The duplication at 15q13.1 overlapped exons of the *Amyloid Beta Precursor Protein Binding Family A Member 2* (*APBA2*) gene, variants in which have been reported in ASD and psychiatric conditions [22]. The 1q43 duplication encompassed the intronic region of the *Phospholipase D Family Member 5* (*PLD5*) gene, variants in which have not been reported in ASD. Both CNVs in DM were classified as VUS by the research laboratory. Genetic counseling was recommended for the siblings as the genetic results had a possible link to their neurodevelopmental condition and health implications for them and their families. The genetic findings had relevance to their current and future healthcare and necessitated disclosure.

The research team contacted the family to obtain their preference for accessing the information (Principle 2). With the family’s permission, the MRC (pediatrician) was contacted. The MRC facilitated confirmatory clinical genetic testing for DM (clinical microarray was already available for NM) and referred the family to the regional clinical genetic clinic. The case highlights how a collaborative approach to RoR, by implicating the MRC, facilitated clinical integration of research genetic results and access to routine clinical care (Principle 3 and 4).

### 3.4. Case 15

At enrollment, TZ was a teenager with Asperger’s Syndrome, enrolled with his family in the genetic study. Research microarray showed a maternally inherited deletion at Xp22.31, affecting several genes, including *Steroid Sulfatase* (*STS*). Mutations in *STS* have been associated with X-linked ichthyosis (MIM 308100) [23]. Affected individuals may have extra-cutaneous manifestations [24], ASD, and other neurodevelopmental conditions [23]. The research laboratory classified the result as likely pathogenic and genetic counseling was recommended. The genetic finding was relevant to TZ’s neurodevelopmental diagnosis and other health aspects, so it required communication.

TZ had reached adulthood since enrollment, which took place seven years prior to the availability of the genetic result. The research site notified the family that a research result was available. TZ was an adult capable of making personal health decisions. His mother was also a participant in the study and a carrier of the genetic change. Both had the opportunity to independently express their preferences to learn about the result. TZ had never had clinical genetic testing. He chose not to pursue the matter further. However, his mother expressed interest in learning about the genetic result and its implications for her and TZ’s unaffected sibling. With the mother’s permission, the research team contacted the MRC (family doctor) and collaborated on the care coordination for the mother and sibling. The MRC referred them to the genetic clinic in their healthcare jurisdiction for confirmatory testing and counseling. This case underscores the importance of eliciting the expectations and preferences of research participants for whom a research genetic result is available (Principle 2). The personal choice about the receipt of results may differ, even among family members. Genetic results may impact family members differently, based on several factors, including carrier status and clinical profile (Principle 4). 

## 4. Discussion

The interpretation of genetic findings in ASD requires careful consideration of existing genetic information in a highly individualized context. Current clinical RoR recommendations do not offer ASD-specific guidance and lag behind novel information from genomic studies. Within research, various RoR recommendations focus on specific topics, but do not present a practical roadmap for the return of genetic findings to participants in ASD research. Our case series illustrate some of the complexity of RoR from genetic research studies in ASD, although it certainly does not capture the breadth of potential challenges that may be encountered during RoR. We demonstrate that RoR entails an overlap of ethical issues, complex science, clinical considerations, and health systems (Figure 1). We utilized a framework of principles derived from the literature (Table 1) in order to provide systematic guidance on RoR from research, in order to resolve potential challenges and integrate actionable genetic research results into clinical care pathways for participants. This approach facilitates a mutually beneficial partnership between clinical and research domains. The framework we propose may serve to steer the RoR process and the clinical integration of genetic research findings in ASD. 

### 4.1. Principle 1: Relevance of Genetic Result to Current and Future Care

Genomic discovery research casts a wide net in order to capture the numerous genes involved in brain development and function, maximizing the chance of actionable findings. Existing ethics recommendations state that researchers must outline if and how their expected genetic results will be returned to participants [11,12,25,26]. Return of *actionable* genetic findings is now accepted as standard ethical conduct and research teams are encouraged to outline an RoR plan as part of their ethics-approved research protocol [11,25,26,27,28]. However, interpreting the actionability of genetic findings can be challenging [7]. Recommendations for the interpretation and reporting of results from *clinical* genetic tests have offered broad classification [6,29,30]. However, the onus is on clinical laboratories and geneticists to make an informed decision about the actionability of the results. The actionability of genetic results is a moving target that is clinically defined based on advances made in research demonstrating associations of specific variants in individuals with ASD or related conditions. In research, clinical guidelines are even less applicable, as they do not account for novel information generated by advanced genetic tests. We suggest that research teams consider the available scientific literature and expert opinion regarding the identified genetic finding, together with the participants’ personal health factors in determining or confirming the actionability of a research result. This may be achieved by collecting detailed health data for a contextual interpretation of the genetic research result. In some cases, participants may already have knowledge of the genetic finding through clinical genetic testing available to them, which may eliminate the need for RoR from research, as was the case with one of our participants (Table 2, Case 10). Thus, during the RoR process, researchers should synthesize genetic and personal information to determine the actionability and the need for RoR of research findings.

### 4.2. Principle 2: Participant/Family Expectations, Preferences and Decisions

Ethics recommendations favor clear communication at the time of consent of the researcher’s plan for returning results [12,26]. Adult participants should be offered the choice to opt-out of receiving personal genetic results [11,28]. A study showed that although most participants valued receiving incidental findings from research, personal utility depended on the type of finding and not all participants wanted to receive results [31]. This suggests that a “one-size-fits-all” approach in RoR is not ideal. The timing of RoR also influences personal utility. A study of the return of actionable results to cancer research participants showed that timing of RoR within the individual’s current life experiences was important [32]. The perspectives of the person/family receiving the results must be considered, which are modified by time and their healthcare journey. Consent should be an ongoing process [12], which is especially important given the time lag between research consent and RoR. The participant’s decision at consent about the return of research results should be confirmed before RoR [12]. This also ensures that the preferences of adult participants, enrolled as children, are taken into consideration.

### 4.3. Principle 3. Person/Family-Centeredness

Effective RoR from research relies on the active engagement of routine health services to ensure that resulting care pathways are personalized, clear, and accessible. This at a minimum includes confirmatory testing of the research result at a certified clinical laboratory [28] and the involvement of genetic specialists. The integration of genetic research results into clinical care raises the issue of impact on health system resources. Concerns have been expressed about the practicality, infrastructure, and costs of such integration [33]. The integration of genetic research results in the healthcare of participants is not meant to replace clinical genetic testing. It is meant to enhance the participant’s healthcare if an actionable genetic research result is identified. Moreover, technologies used in genomic research may soon become the standard of clinical care [34]. Thus, RoR from research can provide valuable insights into the integration of complex genetic information in personal healthcare. This knowledge may help with the implementation of more powerful genetic tests in clinical care for neurodevelopmental conditions.

### 4.4. Principle 4. Care Coordination

Collaboration between the research team and the MRC can foster person/family-centered healthcare pathways from the return of genetic research results. Primary care providers desire increased knowledge, closer ties to genetics specialists, and access to reliable resources about personalized medicine [35]. Models of RoR where the research team takes on aspects of clinical care, like genetic counseling, run in parallel to existing healthcare pathways. They may not address all of the individual’s needs and may lack longitudinal involvement. We propose that the research team collaborate with healthcare providers on the integration of genetic research results into clinical care. The MRC can support the individual/family through the receipt of genetic results and ensure that they navigate relevant services in the context of their specific healthcare setting or jurisdiction. A collaborative model between research teams, healthcare providers, and genetic specialists offer continuity and person/family centered care.

### 4.5. Principle 5: Benefits and Risks

Qualitative studies suggest that genetic results can have both positive and negative effects on families. For example, genetic results may inform the prognosis, medical management, and health surveillance of a child [36,37]. Some parents report a sense of comfort in knowing the biological cause of their child’s condition, and that genetic results improve access to services [1,36,37]. On the other hand, parents may have high levels of uncertainty about the meaning of the genetic result for their child or family [37,38]. They may have negative emotional responses, like guilt or blame, about the heritability of a genetic finding, and disappointment with the lack of meaningful impact on services [36,37,38]. We suggest that there should be an effort by research teams to consider the potential positive and negative impacts for each participant when making decisions about aspects of RoR (such as timing, mode, and ensuing consequences). An important goal should be to actively monitor and minimize negative effects on participants receiving genetic research results, in collaboration with the MRC and/or clinical genetic services if required. 

The proposed framework for RoR (Figure 1) offers guidance on returning actionable genetic results to research participants (when such results are identified by the research team) and integrating them into individual clinical care. It serves as a scaffold for a systematic approach to RoR, aiming to bridge the gap between research and healthcare. The framework should be applied in an individualized and flexible manner, adapted to existing healthcare systems and the needs of participants. The goal is to maximize the application of genetic knowledge in the care of individuals on the autism spectrum, through a tailored and collaborative process at the level of the individual and their health system. We acknowledge that for many research participants, no actionable results may be found, or identified genetic results may not have significant clinical utility. The identification of a genetic etiology in a participant may have implications for family members who are carriers, but who are unaffected or have traits of the broader ASD phenotype. Receipt of genetic information may also have implications for the personal utility of participants, even if the genetic result has no measurable impact on clinical care. We believe that our RoR framework inherently addresses the issue of scaling up to accommodate the greater number of actionable genetic findings that will be generated from sequencing technologies used in research, because it relies on an integration within existing clinical pathways, rather than the creation of new infrastructures. In addition, it can be adapted for single or multiple time point disclosures of genetic findings, to account for the reinterpretation of research findings or reclassification of variants, as genomic technologies and scientific knowledge evolve. The RoR framework may apply to contexts beyond ASD, such as intellectual disability, given the genetic overlap between neurodevelopmental conditions. Further work should focus on providing greater guidance to research teams on determining the actionability of a research finding, prior to initiating RoR, as actionability determination is a moving target due to the rapidly evolving genetic knowledge in ASD and related conditions. This will also help guide the application of the RoR process as genomic technologies evolve in the future. Research is also needed to further test and refine our proposed RoR framework in larger samples. Lastly, it may be informative to further understand the impact of RoR on the individual and broader system level, such as through patient reported outcome measures, measures of clinical and personal utility, and assessments of resource and economic impact. This will allow the refinement of the RoR process as genetic discovery research continues to enhance our understanding of neurodevelopmental conditions.

## Figures and Tables

**Figure 1 jpm-11-00755-f001:**
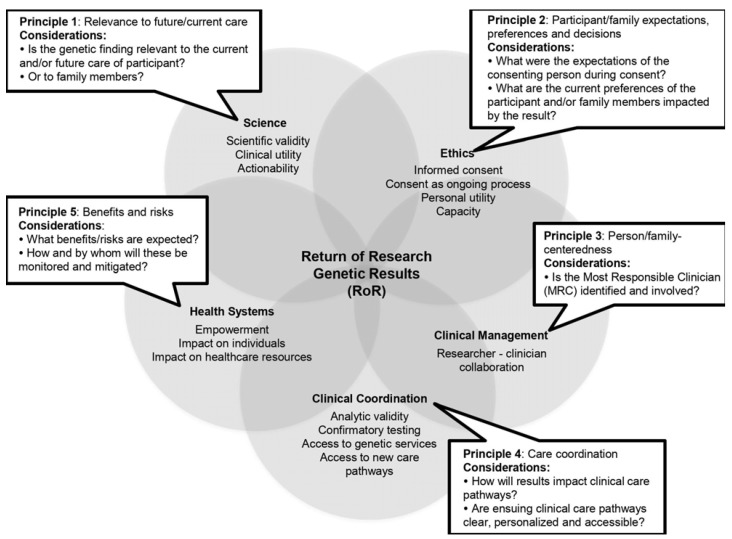
Summary of the proposed RoR framework, its principles, and their alignment within ethical, scientific, and healthcare domains. We offer a framework for RoR of actionable genetic research results that can guide research teams on specific points they should consider during the RoR process, with the aim of integration of genetic research results into the routine care of participants. The RoR framework was created through an iterative process that included review of the existing literature on topics related to RoR and review of real-life cases necessitating RoR from genetic research by a multi-disciplinary expert work group at our center. The proposed RoR framework entails five guiding principles that are centered within ethical, scientific, clinical management, clinical coordination and health system aspects of RoR. Within each of these domains, we highlight specific topics identified in the existing RoR literature that merit consideration by research teams engaged in RoR. We suggest that optimal use of genetic research results relies on their integration into individualized care pathways for participants.

**Table 1 jpm-11-00755-t001:** Summary of existing recommendations and discussions on the return of genetic results.

RoR Theme	Applicable Questions	Relevant References
RoR process	Formulated with aid from an independent advisory committee?	Fabsitz et al., 2010
	Explicitly stated in the study protocol approved by Ethics Board?	Caulfield et al., 2008
		Miller et al., 2010
		Pres. Commission. 2013
		Tri-council Policy 2014
		Sénécal et al., 2015
		Thorogood et al., 2019
	Consistent with legal and ethical frameworks?	Fabstitz et al., 2010
		Wolf et al., 2012
		Zawati et al., 2014
		Thorogood et al., 2019
	Current and future specific tests (e.g., microarray, WES, WGS) characteristics considered?	Fabstitz et al., 2010
		Zawati et al., 2014
		Thorogood et al., 2019
		Malinowski et al., 2020
	Family context considered?	Knoppers et al., 2013
		Zawati et al., 2014
		Sénécal et al., 2015
		Thorogood et al., 2019
	Management of incidental/secondary findings considered?	Wolf et al., 2008
		Pres. Commission. 2013
		Green et al., 2013
		Tri-council Policy 2014
		Thorogood et al., 2019
	Expertise available to aid result interpretation?	Caulfield et al., 2008
		Wolf et al., 2012
		Green et al., 2013
		Tri-council Policy 2014
		Holm et al., 2014
		Zawati et al., 2014
		Sénécal et al., 2015
Individual preferences	Preferences for RoR of individual?	Caulfield et al., 2008
		Wolf et al., 2008
		Fabsitz et al., 2010
		Wolf et al., 2012
		Green et al., 2013
		Knoppers et al., 2013
		Pres. Commission 2013
		Tri-council Policy 2014
		Jarvik et al., 2014
		Holm et al., 2014
		Zawati et al., 2014
		Sénécal et al., 2015
		Thorogood et al., 2019
	Process of RoR for minors (whose guardians are consented)?	Wolf et al., 2008
		Green et al., 2013
		Jarvik et al., 2014
		Zawati et al., 2014
		Sénécal et al., 2015
	Preferences for re-contact for results and/or further studies?	Wolf et al., 2008
		Thorogood et al., 2019
	Involvement of Most Responsible Clinician in RoR process?	Wolf et al., 2008
		Sénécal et al., 2015
Criteria for RoR in individual cases	Is the finding primary, secondary or incidental?	Wolf et al., 2008
		Pres. Commission.2013
		Green et al., 2013
		Thorogood et al., 2019
	Does it have current and/or future health implication?	Caulfield et al., 2008
		Wolf et al., 2008
		Fabsitz et al., 2010
		Green et al., 2013
		Knoppers et al., 2013
		Sénécal et al., 2015
		Thorogood et al., 2019
	Is it clinically actionable?	Caulfield et al., 2008
		Fabsitz et al., 2010
		Wolf et al., 2012
		Green et al., 2013
		Knoppers et al., 2013
		Jarvik et al., 2014
		Sénécal et al., 2015
	Does it have therapeutic benefit?	Caulfield et al., 2008
		Wolf et al., 2008
		Fabsitz et al., 2010
		Green et al., 2013
		Knoppers et al., 2013
		Jarvik et al., 2014
		Sénécal et al., 2015
	Is it analytically valid?	Caulfield et al., 2008
		Wolf et al., 2008
		Fabsitz et al., 2010
		Wolf et al., 2012
		Knoppers et al., 2013
		Jarvik et al., 2014
		Holm et al., 2014
		Sénécal et al., 2015
		Thorogood et al., 2019

**Table 2 jpm-11-00755-t002:** Characteristics and genetic findings of participants for whom a genetic research result was available.

Case	Biological Sex	Affected Region	Type	Inheritance	Clinical Significance	Age at RoR (Years)	Outcome of RoR
1	M	hg19 chr1: g.[5663T>G]	SNV	De novo	*mTOR* involvement	6	Result returned Clinical care provided
2	F	hg19 chr2: g.[230701696G>A]	SNV	De novo	*TRIP12* involvement Nonsense mutation	24	Result returned Clinical care provided
3	M	hg19 chr10: g.[89692908C>T]	SNV	De novo	*PTEN* involvement Known missense effect Characterized syndrome	10	Result returned Clinical care provided
4	F	hg19 chr16: g.[2131695C>T]	SNV	De novo	*TSC2* involvement Missense mutation Characterized syndrome	19	Result returned Clinical care provided
5	M	1q21.1	CNV dup	De novo	1.4 Mb del. of 10 genes Characterized syndrome	14	Result returned Clinical care provided
6	M	1q21.1	CNV dup	De novo	1.4 Mb del. of 10 genes Characterized syndrome	15	Result returned Clinical care provided
7	M	9q21.13	CNV del	De novo	4.8 Mb del. of 18 genes	12	Lost to follow up
8	M	15q11.2	CNV del	Maternal	VUS 512.4 kb del. of 4 genes	11	Result returned Clinical care provided
9	M	15q13.1 1q43	CNV dup CNV dup	Maternal De novo	VUS 254 kb dup. in 1 gene VUS 28.6 kb dup. in 1 gene	12	Result returned Clinical care provided
10	M	15q13.2	CNV del	Unknown	1.59 Mb del. of 5 genes Characterized syndrome	10	Results previously identified on clinical genetic testing
11	M	16p11.2	CNV dup	Paternal	561 kb dup. of 30 genes Characterized syndrome	20	Lost to follow up
12	M	16p11.2	CNV dup	De novo	633 kb dup. of 31 genes Characterized syndrome	12	Result returned Clinical care provided
13	M	16p13.11	CNV del	Paternal	1.2 Mb del. of 13 genes Characterized syndrome	14	Result returned Clinical care provided
14	M	16p13.11	CNV del	De novo	921 kb del. of 9 genes Characterized syndrome	20	Result returned Clinical care provided
15	M	Xp22.31	CNV del	Maternal	1.6 Mb del. of 5 genes Characterized syndrome	22	Result returned Clinical care provided
16	M	XXY	Aneu-ploidy	De novo	Characterized syndrome	22	Lost to follow up

## Data Availability

The data presented in this study are available on request from the corresponding author.

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
