# Peer review of "Enhancing the Impact of Genomics Research in Autism through Integration of Research Results into Routine Care Pathways—A Case Series"

_jpm, 2021, doi:10.3390/jpm11080755_

Round 1
Reviewer 1 Report
Your work represents an important framework for practitioners in the field. I have one important methodological clarification as well as a 2 ACMG points to consider which may be relevant to your article.
In the methods, so do not discuss the circumstances under which assent was sought from minors. Please clarify the circumstances under which assent was sought, both in the original studies as well as in your RoR framework
2 ACMG Points to Consider/guidelines:
1) Recent "Systematic evidence-based review: outcomes from exome and genome sequencing for pediatric patients with congenital anomalies or intellectual disability" from ACMG means that, in some areas, the technologies from the studies are now essentially part of the standard of care. At my institution, for example, patients with ID/ASD essentially all get an array and an exome. The only research RoR is new gene discovery. Your paper addresses mostly finding known clinical entities because the subjects clinicians were not using genomic technologies on a clinical basis. Your manuscript would benefit from at least considering how your framework might apply in the near future as technologies evolve.
2) ACMG also has a points to consider about recontact at https://www.nature.com/articles/s41436-018-0391-z which is obviously primarily for clinical labs, but it is interesting that your paper deals with single time point disclosures. Again, your manuscript would be strengthened by at least considering how your proposed framework integrates with the possible reinterpretation of research findings/reclassification of variants.
Author Response
Dear Reviewer 1 and Editors:
Thank you for the insightful feedback. Here are our responses to your comments:
Point 1. Could the authors give details of the total number of participants in the research studies as a denominator.
Response 1. The Reviewer makes a good observation that the total number of returned results was small. The total number of participants through our site that enrolled in the large-scale genomic studies outlined in our paper (namely, Simons Simplex Collection and the MSSNG database) is approximately 400. The number of actionable research results in our study is small for two main reasons:
- At the time of initial analysis (spanning 2007-2017), both studies were using microarrays, whose yield is limited. It was only towards the end of that period that they began returning results from exome sequencing. In fact, these studies are now re-analysing all samples using exome and genome sequencing, which is already resulting in greater number of identified actionable results. Our site continues to undergo the RoR process with other participants as more actionable findings are identified.
- The main research sites use very conservative criteria for identifying actionable results, as they relate to ASD, which likely also impact the ‘diagnostic’ yield.
Point 2. The discussion in terms of practicalities would be strengthened if the problems of scaling up this intensive approach could be realised. For example in large scale sequencing projects one would expect many more findings to have to be returned and it would be helpful to have an idea of how scalable the authors thought that such an approach might be.
Response 2. The Reviewer points out an important consideration. It is true that sequencing technologies will yield greater number of actionable genetic findings from research. We believe that our proposed RoR Framework inherently addresses the question of scaling up, because it relies on an integration within existing clinical pathways (i.e. the participants’ existing primary care providers and clinical genetic services), rather than the creation of new infrastructures. We also believe that existing clinical genetic services are already expanding to meet the growing demands of clinical systems as sequencing technologies are entering the clinical realm, not just for ASD but for other conditions as well. We have made some additions to our Discussion, in order to address this issue (pg. 15, line 46; pg. 16, line 1-3).
Point 3. Access to health care varies in different jurisdictions. For the participants, was access to funded health care guaranteed. This is also an important factor in considering return of results into a health care systems, that affects the generalisability of the framework.
Response 3. We agree with the Reviewer that generalisability of our RoR Framework to some extent depends on the healthcare system and jurisdiction. Our study site is in Montreal, Canada. In Canada, there is a publicly funded health care system that guarantees access to a primary care provider and specialty clinics, like genetic clinics. The strength of our proposed RoR framework is that it provides general principles that should be tailored to the individual healthcare system and participant. We have added a clarifying statement to highlight that the RoR framework needs to be adapted to the existing healthcare system (pg. 15, lines 37-38).
Thank you for your consideration of our revised manuscript.
Sincerely,
Iskra Peltekova, MD, MSc (corresponding author)
Holland Bloorview Kids Rehabilitation Hospital, Toronto, Ontario, Canada
ipeltekova@hollandbloorview.ca

Reviewer 2 Report
Thank for asking me to review this interesting paper which described a framework for return of results. The overall conclusion that 'optimal use of genetic research results relies on their integration into individualised care pathways for participants' is certainly one I profoundly agree with and it does need stating. The development of the framework was underpinned by literature and is very relevant and practical. However my criticism of this paper is that the number of participants described was small, only 16, and this was over a number of years. My specific comments are :
Could the authors give details of the total number of participants in the research studies as a denominator.
The discussion in terms of practicalities would be strengthened if the problems of scaling up this intensive approach could be realised. For example in large scale sequencing projects one would expect many more findings to have to be returned and it would be helpful to have an idea of how scalable the authors thought that such an approach might be.
Access to health care varies in different jurisdictions. For the participants, was access to funded health care guaranteed. This is also an important factor in considering return of results into a health care systems, that affects the generalisability of the framework.
Author Response
Dear Reviewer 2 and Editors:
Thank you for the insightful feedback. Here are our responses to your comments:
Point 1. In the methods, so do not discuss the circumstances under which assent was sought from minors. Please clarify the circumstances under which assent was sought, both in the original studies as well as in your RoR framework.
Response 1. The Reviewer raises an important point. At time of enrollment in the original studies, a research assistant obtained consent from a parent/legal guardian and assent from any participating minor who was able to provide it (as outlined in the Ethics protocol for the studies). During the RoR process, the research team explored the preferences for receipt of the results from the parent/legal guardian who provided the initial consent, as well as from the participants who were adults by the time results became available. The process of exploring current preferences of participants is in fact part of the RoR Framework (Principle 2 on pg. 6, line 4) We have added a clarifying statement to our Methods section (pg. 5, line 14-16).
Point 2. Recent "Systematic evidence-based review: outcomes from exome and genome sequencing for pediatric patients with congenital anomalies or intellectual disability" from ACMG means that, in some areas, the technologies from the studies are now essentially part of the standard of care. At my institution, for example, patients with ID/ASD essentially all get an array and an exome. The only research RoR is new gene discovery. Your paper addresses mostly finding known clinical entities because the subjects clinicians were not using genomic technologies on a clinical basis. Your manuscript would benefit from at least considering how your framework might apply in the near future as technologies evolve.
Response 2. The Reviewer makes a very important observation about the distinctions between different healthcare systems and jurisdictions. In the healthcare system where our research site is located (which is a publicly funded healthcare system), at present, individuals with ASD and ID receive only a clinical microarray and Fragile X analysis as screening clinical genetic tests. Exome and genome sequencing are used on a case-by-case basis based on specific clinical and administrative criteria, as set out by our provincial body that oversees healthcare. As we outline in one of our responses to Reviewer 1, the strength of our proposed RoR framework is that it provides general principles that can be tailored to the individual healthcare system and participant. We have added a clarifying statement to highlight that the RoR framework needs to be adapted to the existing healthcare system and technologies (pg. 15, lines 37-38). We have also added a statement highlighting the need for greater guidance to research teams in determining the actionability of novel variants, which will also help guide the application of the RoR framework as genomic technologies evolve (pg. 16, lines 8-9). We have also added the suggested ACMG point to our References (Reference #2).
Point 3. ACMG also has a points to consider about recontact at https://www.nature.com/articles/s41436- 018-0391-z which is obviously primarily for clinical labs, but it is interesting that your paper deals with single time point disclosures. Again, your manuscript would be strengthened by at least considering how your proposed framework integrates with the possible reinterpretation of research findings/reclassification of variants.
Response 3. We agree with the Reviewer that reinterpretation/reclassification of findings will impact the RoR process from research. In fact, the concept of re-analysis and reclassification is embedded in the informed consent process that participants underwent at enrollment into the genomic studies for which we serve as a site. That is because for both studies (Simon Simplex Collection and MSSNG database) are still ongoing as research samples are re-analysed using novel technologies. Our RoR Framework provides broad principles that can be used for single or recurring disclosures, as it does not explicitly limit the number of disclosures. We have added a clarifying point to address this concept (pg. 16, lines 3-6). We have also added this ACMG point to our references (Reference #14).
Thank you for your consideration of our revised manuscript.
Sincerely,
Iskra Peltekova, MD, MSc (corresponding author)
Holland Bloorview Kids Rehabilitation Hospital, Toronto, Ontario, Canada
ipeltekova@hollandbloorview.ca
